

# The Bayesian confidence intervals for measuring the difference between dispersions of rainfall in Thailand

Noppadon Yosboonruang, Sa-Aat Niwitpong and Suparat Niwitpong

Department of Applied Statistics, Faculty of Applied Science, King Mongkut's University of Technology North Bangkok, Bangkok, Thailand

## ABSTRACT

The coefficient of variation is often used to illustrate the variability of precipitation. Moreover, the difference of two independent coefficients of variation can describe the dissimilarity of rainfall from two areas or times. Several researches reported that the rainfall data has a delta-lognormal distribution. To estimate the dynamics of precipitation, confidence interval construction is another method of effectively statistical inference for the rainfall data. In this study, we propose confidence intervals for the difference of two independent coefficients of variation for two delta-lognormal distributions using the concept that include the fiducial generalized confidence interval, the Bayesian methods, and the standard bootstrap.

The performance of the proposed methods was gauged in terms of the coverage probabilities and the expected lengths via Monte Carlo simulations. Simulation studies shown that the highest posterior density Bayesian using the Jeffreys' Rule prior outperformed other methods in virtually cases except for the cases of large variance, for which the standard bootstrap was the best. The rainfall series from Songkhla, Thailand are used to illustrate the proposed confidence intervals.

## INTRODUCTION

Recently, the Earth's climate has been changing significantly due to the greenhouse effect, which is causing both rising temperatures and variability in precipitation (*Attavanich, 2013*). In particular, Thailand, which is an agricultural country, is greatly affected by such phenomena since agriculture mainly relies on rainfall. The amount of rainfall in Thailand fluctuates quite widely due to the influence of the southwest and northeast monsoons (*Eso, Kuning & Chuai-Aree, 2015*). In previous years, several areas in Thailand have been affected by heavy rain that produced flooding, a major cause of economic, life, and property loss.

It is important to investigate the coefficient of variation of rainfall data series to understand the dynamics of precipitation in each area. Furthermore, the difference between two areas or time periods of heavy rainfall measured with their coefficients of variation is of interest. The government can use this information for advanced planning to

Corresponding author
Suparat Niwitpong, suparat.n@sci.kmutnb.ac.th

prevent problems caused by excessive rainfall. Many researchers have found that rainfall data series follow a bivariate lognormal distribution (a delta-lognormal distribution) (*Fukuchi, 1988*; *Shimizu, 1993*; *Kong et al., 2012*; *Maneerat, Niwitpong & Niwitpong, 2019a*, *2019b*; *Yosboonruang, Niwitpong & Niwitpong, 2019b*; *Yue, 2000*).

Confidence interval construction is another method of effective statistical inference for applying to delta-lognormal distributions and methods to construct them have been reported by several researchers. *Zhou & Tu (2000)* proposed confidence intervals for the mean including a percentile-*t* bootstrap interval based on sufficient statistics, a bias-corrected maximum likelihood method, and an interval based on a likelihood ratio testing method; the bootstrap interval performed the best for both one-sided and two-sided intervals with a small sample size. *Tian (2005)* compared the generalized variables method and the generalized pivotal quantity (GPQ) to construct confidence intervals for the mean, between which the generalized variables method was preferable. *Tian & Wu (2006)* recommended using the adjusted signed log-likelihood ratio statistic to construct confidence intervals for the mean. *Chen & Zhou (2006)* considered interval estimations for the ratio of or difference between two means using a true generalized pivotal (GP) method, an approximate GP method, a signed log-likelihood ratio method, and a modified signed log-likelihood ratio method; their results show that the approximate GP method performed the best. *Fletcher (2008)* used three methods, Aitchison's estimator, a modification of Cox's method, and a profile-likelihood interval, to construct confidence intervals for the mean; they found that the profile-likelihood interval was the best unless the sample size was small with a low-to-moderate level of skewness. *Li, Zhou & Tian (2013)* presented an approximate GPQ and the fiducial quantity to establish confidence intervals for the mean; their results indicate that the fiducial method was the most suitable. *Wu & Hsieh (2014)* introduced the generalized confidence interval (GCI) to construct confidence intervals for the mean that were better than Aitchison's method, a modified Land's method, and the profile-likelihood interval. *Maneerat, Niwitpong & Niwitpong (2018)* constructed confidence intervals for the mean using GCI, the method of variance estimate recovery (MOVER) based on the variance stabilizing transformation (VST), Wilson's score, and Jeffrey's method; GCI and the three MOVER methods had similar performances except for cases where the probability had values close to zero and the coefficient of variation was large. Moreover, they compared GCI and MOVER based on a weighted beta distribution and VST to construct confidence intervals for the mean and recommended MOVER based on VST (*Maneerat, Niwitpong & Niwitpong, 2019b*). In addition, *Maneerat, Niwitpong & Niwitpong (2019a)* suggested Bayesian methods to construct the highest posterior density (HPD) intervals for a single mean and the difference between two means.

Apart from the mean, the coefficient of variation, which is defined as the ratio of the standard deviation to the mean, has been used to solve this statistical problem. There have been many studies focused on confidence interval estimation for the coefficient of variation of normal and non-normal distributions. For instance, *Wong & Wu (2002)* constructed confidence intervals by developing small sample asymptotic methods for both normal and non-normal models. In addition, confidence interval estimations for the coefficient of

variation of a normal distribution have been reported by *Tian (2005)*, *Donner & Zou (2012)*, and *Wongkhao, Niwitpong & Niwitpong (2015)*. Confidence intervals for the coefficient of variation have been established for skewed distributions. *Sangnawakij & Niwitpong (2017a)* presented confidence interval estimations for the coefficient of variation and the difference between coefficients of variation based on MOVER, GCI, and asymptotic confidence interval for two-parameter exponential distributions, their results indicating that GCI outperformed the other methods. *Thangjai & Niwitpong (2017)* proposed confidence intervals for the weighted coefficients of variation of two-parameter exponential distributions using the adjusted MOVER, GCI, and large sample methods, their result showing that GCI was the best choice. *Yosboonruang, Niwitpong & Niwitpong (2018)* constructed confidence intervals for the coefficient of variation of a delta-lognormal distribution using GCI and a modified Fletcher method and found that GCI was the most appropriate. *Yosboonruang, Niwitpong & Niwitpong (2019a)* presented the fiducial generalized confidence interval (FGCI) and MOVER to construct confidence intervals for three parameters of a delta-lognormal coefficient of variation. They reported that FGCI was suitable for small sample sizes while MOVER performed similarly well to FGCI when the sample sizes were large. In addition, they constructed confidence intervals using Bayesian methods with equitailed confidence intervals and the HPD interval and compared them with FGCI; their results show that the Bayesian equitailed confidence interval was appropriate in all cases (*Yosboonruang, Niwitpong & Niwitpong, 2019b*).

Confidence interval estimations for functions of the coefficient of variation are of interest. For normal distributions, *Donner & Zou (2012)* presented MOVER to construct a confidence interval for the difference between two coefficients of variation. Their proposed method performed well for both the coverage percentage and balance between the tail errors. *Niwitpong (2015)* proposed confidence intervals for the difference between the coefficients of variation with bounded parameters; their results show that their proposed confidence intervals outperformed other classical ones in terms of the coverage probability and the average length.

For skewed distributions, *Buntao & Niwitpong (2012)* constructed confidence intervals for the difference between coefficients of variation for lognormal and delta-lognormal distributions by using the GP method and a closed-form method of variance estimation; their results for both lognormal and delta-lognormal distributions indicate that the GP method was better than the closed-form method in all cases. *Buntao & Niwitpong (2013)* produced confidence intervals for the ratio of the coefficients of variation of delta-lognormal distributions based on the GP method and the MOVER based Wald interval; they suggested that the GP method was the most appropriate. *Sangnawakij & Niwitpong (2017b)* constructed new confidence intervals for functions of the difference between and the ratio of the coefficients of variation with restricted parameters in two gamma distributions; they found that the expected lengths of the proposed confidence intervals were shorter than other classical estimators.

Although a number of previous studies have reported on constructing confidence intervals for several parameters in each distribution, there has been only one study on

constructing confidence intervals for the difference between the coefficients of variation of two delta-lognormal distributions. Constructing confidence intervals using general methods are quite complex. Furthermore, results have revealed that the performances of these methods are not consistent since the coverage probabilities are less than the target in a few cases. From the perspective of rainfall data, estimating the difference between two independent coefficients of variation can help to elucidate rainfall variability in terms of time or area. It is useful for forecasting rainfall to help in planning for and managing risky situations that can arise from rainfall variation.

In this study, the difference between the coefficients of variation of two delta-lognormal distributions was investigated. In previous studies (*Donner & Zou, 2012*; *Li, Zhou & Tian, 2013*; *Wu & Hsieh, 2014*; *Sangnawakij & Niwitpong, 2017a*; *Thangjai & Niwitpong, 2017*; *Maneerat, Niwitpong & Niwitpong, 2018, 2019a*; *Yosboonruang, Niwitpong & Niwitpong, 2018, 2019a*), confidence intervals for the difference between the coefficients of variation of two delta-lognormal distributions were constructed using three methods (GCI, FGCI, and MOVER). Our preliminary study indicates that these methods performed similarly, although FGCI is the best due to having the shortest expected length. Therefore, we constructed new confidence intervals for the difference between the coefficients of variation of two delta-lognormal distributions using Bayesian methods and a standard bootstrap (SB) method and compared them with FGCI. The details of each method are presented in the next section, after which the results are presented. Next, the efficacies of the proposed methods for constructing confidence intervals are illustrated using rainfall data in an empirical example, followed by a discussion and conclusions of the study outcomes.

## MATERIALS AND METHODS

In statistical inference and its applications, data containing non-negative values can be skewed and many zero observations can be observed. *Aitchison (1955)* introduced the delta-lognormal distribution for data series containing non-negative values and true-zero values of the variables. The positive observed values, denoted by $n_{i(1)}$, have a lognormal distribution, and the true-zero observed values, denoted by $n_{i(0)}$, have a binomial distribution with the probability of zero observations $\tilde{\delta}_i = 1 - \delta_i$, where $n_i = n_{i(1)} + n_{i(0)}$. Let $\mathbf{X}_{ij} = (X_{i1}, X_{i2}, \ldots, X_{in_i})$ be a non-negative random sample from a delta-lognormal distribution with parameters $\delta_i$, $\mu_i$ and $\sigma_i^2$, denoted by $X_{ij} \sim \Delta(\delta_i, \mu_i, \sigma_i^2)$. The distribution function of the delta-lognormal distribution presented by *Tian & Wu (2006)* is

$$F(x_{ij}; \delta_i, \mu_i, \sigma_i^2) = \begin{pmatrix} \tilde{\delta}_i & ; & x_{ij} = 0 \\ \tilde{\delta}_i + \delta_i H(x_{ij}; \mu_i, \sigma_i^2) & ; & x_{ij} > 0, \end{pmatrix} \tag{1}$$

where $H(x_{ij}; \mu_i, \sigma_i^2)$ is the lognormal cumulative distribution function. Assume that $Y_{ij} = \ln(X_{ij})$, $i = 1, 2$, $j = 1, 2, \ldots, n_{i(1)}$ is a normal distribution with mean $\mu_i$ and variance $\sigma_i^2$. Thus, the population mean and variance of a delta-lognormal distribution as presented by *Aitchison (1955)* are $E(X_{ij}) = \delta_i \exp(\mu_i + \sigma_i^2/2)$ and

$Var(X_{ij}) = \delta_i \exp(2\mu_i + \sigma_i^2)\left[\exp(\sigma_i^2) - \delta_i\right]$, respectively. Herein, we focus on confidence interval estimations for the difference between the coefficients of variation of two delta-lognormal distributions. The coefficient of variation of a delta-lognormal distribution can be expressed as

$$\mathrm{CV}(X_{ij}) = \eta_i = \frac{\sqrt{Var(X_{ij})}}{E(X_{ij})} = \left[\frac{\exp(\sigma_i^2) - \delta_i}{\delta_i}\right]^{\frac{1}{2}}. \tag{2}$$

It is easy to find the difference between two independent coefficients of variation:

$$\gamma = \eta_1 - \eta_2 = \left[\frac{\exp(\sigma_1^2) - \delta_1}{\delta_1}\right]^{\frac{1}{2}} - \left[\frac{\exp(\sigma_2^2) - \delta_2}{\delta_2}\right]^{\frac{1}{2}}. \tag{3}$$

**The fiducial generalized confidence interval**

The basic concept of the fiducial distribution was introduced by *Fisher (1930)*. Moreover, statistical inference using fiduciality can be found in several studies (*Dawid & Stone, 1982*; *Aldrich, 2000*; *Hannig et al., 2006*; *Hannig, Iyer & Patterson, 2006*; *Hannig, 2009*; *Hannig & Lee, 2009*). After that, *Li, Zhou & Tian (2013)* proposed the generalized fiducial quantity (GFQ) of a population mean by using the concept of fiducial inference and then constructed confidence intervals for the mean based on the fiducial of a lognormal distribution with excess zeros. Furthermore, *Yosboonruang, Niwitpong & Niwitpong (2019a)* recommended FGCI to construct confidence intervals for the coefficient of variation of a delta-lognormal distribution. From *Hannig (2009)* and *Li, Zhou & Tian (2013)*, the GFQs for $\delta_i$ and $\sigma_i^2$ are

$$T_{\delta_i} \sim \frac{1}{2} Beta(n_{i(1)}, n_{i(0)} + 1) + \frac{1}{2} Beta(n_{i(1)} + 1, n_{i(0)}) \tag{4}$$

and

$$T_{\sigma_i^2} = \frac{(n_{i(1)} - 1)\hat{\sigma}_i^2}{U_i}, \tag{5}$$

respectively, where $U_i \sim \chi_{n_{i(1)}-1}^2$. Next, the GFQ for $\gamma$ can be defined as

$$T_\gamma = T_{\eta_1} - T_{\eta_2} = \left[\frac{\exp\left(T_{\sigma_1^2}\right) - T_{\delta_1}}{T_{\delta_1}}\right]^{\frac{1}{2}} - \left[\frac{\exp\left(T_{\sigma_2^2}\right) - T_{\delta_2}}{T_{\delta_2}}\right]^{\frac{1}{2}}. \tag{6}$$

Therefore, the 100 $(1 - \alpha)$% confidence interval for $\gamma$ is

$$CI_\gamma^{FGCI} = \left[T_{\gamma,(l)}, T_{\gamma,(u)}\right] = \left[T_\gamma(\alpha/2), T_\gamma(1 - \alpha/2)\right], \tag{7}$$

where $T_\gamma(\alpha/2)$ and $T_\gamma(1 - \alpha/2)$ are the 100 $(\alpha/2)$-th and 100 $(1 - \alpha/2)$-th percentiles of the distribution of $T_\gamma$, respectively.

**The algorithm to construct FGCI**

1. Generate datasets $\mathbf{x}_{ij}$, $i = 1,2$, $j = 1,2,\ldots,n_i$ from the delta-lognormal distribution.

2. Generate $Beta\left(n_{i(1)}, n_{i(0)} + 1\right)$ and $Beta\left(n_{i(1)} + 1, n_{i(0)}\right)$.

3. Compute $T_{\delta_i}$, $T_{\sigma_i^2}$, and $T_\gamma$.

4. Repeat steps 2 and 3 5,000 times.

5. Compute the $100(1 - \alpha)\%$ confidence intervals for $\gamma$.

6. Repeat steps 1–5 15,000 times.

## Bayesian methods

A delta-lognormal distribution is a combination of the two distributions mentioned earlier, with unknown parameters comprising $\delta_i$, $\mu_i$ and $\sigma_i^2$, denoted as $\theta = (\delta_i, \mu_i, \sigma_i^2)$. To compare the two population coefficients of variation, the joint likelihood function is expressed as

$$L\left(\theta | \mathbf{x}_{ij}\right) \propto \prod_{i=1}^{2} \left\{ \tilde{\delta}_i^{n_{i(0)}} \delta_i^{n_{i(1)}} \prod_{j=1}^{n_{i(1)}} \frac{1}{\sigma_i} \exp\left[ -\frac{1}{2\sigma_i^2} \left(\ln\left(x_{ij}\right) - \mu_i\right)^2 \right] \right\}. \tag{8}$$

Our approach points toward the difference between two independent coefficients of variation, given as Eq. (3), thus the unknown parameters are $\delta_i$, $\mu_i$ and $\sigma_i^2$, denoted as $\tilde{\theta} = (\delta_1, \mu_1, \sigma_1^2, \delta_2, \mu_2, \sigma_2^2)$. The Fisher information of $\tilde{\theta}$ computed by the second-order derivative of the log-likelihood function which is defined as

$$I\left(\tilde{\theta}\right) = -E\left[\frac{\partial^2 \ln(L)}{\partial \tilde{\theta}^2}\right]. \tag{9}$$

By Eq. (8), the Fisher information matrix for $\tilde{\theta}$ becomes

$$I\left(\tilde{\theta}\right) = \text{diag}\left[ \frac{n_1}{\tilde{\delta}_1 \delta_1} \quad \frac{n_1 \delta_1}{\sigma_1^2} \quad \frac{n_1 \delta_1}{2(\sigma_1^2)^2} \quad \frac{n_2}{\tilde{\delta}_2 \delta_2} \quad \frac{n_2 \delta_2}{\sigma_2^2} \quad \frac{n_2 \delta_2}{2(\sigma_2^2)^2} \right]. \tag{10}$$

To establish confidence intervals using the Bayesian methods, the left-invariant Jeffreys, the Jeffreys' Rule, and uniform priors were used. In this study, we are interested in constructing the HPD intervals. The probability of the shortest interval is discovered when the posterior density value at the lower and upper limits is equal, thus the upper and lower tail areas are not necessarily equal (*Bolstad & Curran, 2017*).

### The Bayesian method using the left-invariant Jeffreys prior

Rainfall series that consist of zero and non-zero values follow a combination of two distributions: binomial and lognormal. As mentioned previously, the parameter of interest for a binomial distribution is $\tilde{\delta}_i$ and by using the Fisher information matrix of $\tilde{\delta}_i$, we can obtain the invariant Jeffreys prior by the square root of the determinant of Fisher information matrix which is defined as

$$p\left(\tilde{\delta}_i\right) \propto \sqrt{\left|I\left(\tilde{\delta}_i\right)\right|} \propto \tilde{\delta}_i^{-\frac{1}{2}} \delta_i^{-\frac{1}{2}}, \tag{11}$$

which is *Beta* (1/2,1/2). Subsequently, the posterior distribution of $\tilde{\delta}_i$ for binomial distribution can be expressed as

$$
\begin{aligned}
p\big(\tilde{\delta}_i|n_{i(0)}\big) \quad &\propto \frac{L\big(\tilde{\delta}_i\big)p\big(\tilde{\delta}_i\big)}{\int_{-\infty}^{\infty} L\big(\tilde{\delta}_i\big)p\big(\tilde{\delta}_i\big)d\tilde{\delta}_i} \\
&\propto \tilde{\delta}_i^{n_{i(0)}-\frac{1}{2}}\tilde{\delta}_i^{n_{i(1)}-\frac{1}{2}},
\end{aligned}
\tag{12}
$$

which is *Beta* $(n_{i(0)} + 1/2, n_{i(1)} + 1/2)$. By Eq. (10), the left-invariant Jeffreys prior for the parameter of interest, $\sigma_i^2$, from a lognormal distribution obtained by the square root of the determinant of Fisher information matrix is $p(\sigma_i^2) = 1/\sigma_i^2$ (*Rao & D'Cunha, 2016*). Suppose that $\tilde{\delta}_i$ and $\sigma_i^2$ are independent, then the prior distribution for a delta-lognormal distribution can be written as $p(\tilde{\delta}_i, \sigma_i^2) \propto \sigma_i^{-2}\tilde{\delta}_i^{-\frac{1}{2}}\delta_i^{-\frac{1}{2}}$. Consequently, the joint posterior density function can be defined as

$$
\begin{aligned}
p\big(\tilde{\theta}|\text{data}\big) = \quad \prod_{i=1}^{2} &\left\{ \frac{1}{Beta(a,b)}\tilde{\delta}_i^{a-1}\delta_i^{b-1}\frac{1}{\sqrt{2\pi}\frac{\sigma_i}{\sqrt{n_{i(1)}}}}\exp\left[-\frac{1}{2\frac{\sigma_i^2}{n_{i(1)}}}(\mu_i - \hat{\mu}_i)^2\right] \right. \\
&\left. \times \frac{s^r}{\Gamma(r)}(\sigma_i^2)^{-(r+1)}\exp\left(-\frac{s}{\sigma_i^2}\right)\right\},
\end{aligned}
\tag{13}
$$

where $a = n_{i(0)} + 1/2$, $b = n_{i(1)} + 1/2$, $r = (n_{i(1)} - 1)/2$, $s = (n_{i(1)} - 1)\hat{\sigma}_i^2/2$, $\hat{\mu}_i = \sum_{j=1}^{n_{i(1)}} \ln(x_{ij})/n_{i(1)}$, and $\hat{\sigma}_i^2 = \sum_{j=1}^{n_{i(1)}} [\ln(x_{ij}) - \hat{\mu}_i]^2/(n_{i(1)} - 1)$. Therefore, the posterior distribution of $\tilde{\delta}_i$ is a beta distribution, $\tilde{\delta}_i|\text{data} \sim Beta(n_{i(0)} + 1/2, n_{i(1)} + 1/2)$. Similarly, the posterior distribution of $\sigma_i^2$ is an inverse gamma distribution, $\sigma_i^2|\text{data} \sim Inv - Gamma[(n_{i(1)} - 1)/2, (n_{i(1)} - 1)\hat{\sigma}_i^2/2]$.

### The Bayesian method using the Jeffreys' Rule prior

Based on the Fisher information, the Jeffreys' Rule prior can be obtained from $|I(\theta)|^{\frac{1}{2}}$. Thus, the Jeffreys' Rule priors for $\tilde{\delta}_i$ in a binomial distribution and $\sigma_i^2$ in a lognormal distribution are $p(\tilde{\delta}_i) \propto \tilde{\delta}_i^{-\frac{1}{2}}\delta_i^{\frac{1}{2}}$ and $p(\sigma_i^2) \propto \sigma_i^{-3}$ (*Harvey & Van der Merwe, 2012*), respectively. Following Eq. (2), the parameters of interest are $\tilde{\delta}_i$ and $\sigma_i^2$, which are independent. Thus, the Jeffreys' Rule prior for $(\tilde{\delta}_i, \sigma_i^2)$ of a delta-lognormal distribution can be written as $p(\tilde{\delta}_i, \sigma_i^2) \propto \sigma_i^{-3}\tilde{\delta}_i^{-\frac{1}{2}}\delta_i^{\frac{1}{2}}$. Subsequently, the joint posterior density function is defined as Eq. (13) with $a = n_{i(0)} + 1/2$, $b = n_{i(1)} + 3/2$, $r = n_{i(1)}/2$, and $s = n_{i(1)}\hat{\sigma}_i^2/2$. This leads to the posterior density of $\tilde{\delta}_i$ and $\sigma_i^2$, which follow a beta distribution, $\tilde{\delta}_i|\text{data} \sim Beta(n_{i(0)} + 1/2, n_{i(1)} + 3/2)$, and an inverse gamma distribution, $\sigma_i^2|\text{data} \sim Inv - Gamma(n_{i(1)}/2, n_{i(1)}\hat{\sigma}_i^2/2)$, respectively.

### The Bayesian method using the uniform prior

The uniform priors for $\tilde{\delta}_i$ of a binomial distribution and $\sigma_i^2$ of a lognormal distribution are $p(\tilde{\delta}_i) \propto 1$ (*Bolstad & Curran, 2017*) and $p(\sigma_i^2) \propto 1$ (*Kalkur & Rao, 2017*),

respectively. Since $\tilde{\delta}_i$ and $\sigma_i^2$ are independent, then the uniform prior of a delta-lognormal distribution is $p(\tilde{\delta}_i, \sigma_i^2) \propto 1$. Thus, the joint posterior distribution corresponds with Eq. (13) when $a = n_{i(0)} + 1$, $b = n_{i(1)} + 1$, $r = (n_{i(1)} - 2)/2$, and $s = (n_{i(1)} - 2)\hat{\sigma}_i^2/2$. Subsequently, the posterior distributions of $\tilde{\delta}_i$ and $\sigma_i^2$ are a beta distribution, $\tilde{\delta}_i|\text{data} \sim Beta(n_{i(0)} + 1, n_{i(1)} + 1)$, and an inverse gamma distribution, $\sigma_i^2|\text{data} \sim Inv - Gamma[(n_{i(1)} - 2)/2, (n_{i(1)} - 2)\hat{\sigma}_i^2/2]$, respectively.

Subsequently, the Bayesian HPD intervals are constructed by substituting $\tilde{\delta}_i|\text{data}$ and $\sigma_i^2|\text{data}$ from each method into Eq. (3). The following algorithm was constructed to obtain the $100(1 - \alpha)\%$ HPD intervals for $\gamma$.

**The algorithm to construct the Bayesian HPD intervals**

1. Generate datasets $\mathbf{x}_{ij}$, $i = 1,2$, $j = 1,2,\ldots,n_i$ from a delta-lognormal distribution.
2. Generate the posterior densities of the $\tilde{\delta}_i|\text{data}$.

- *Beta* $(n_{i(0)} + 1/2, n_{i(1)} + 1/2)$ for the left-invariant Jeffreys prior.

- *Beta* $(n_{i(0)} + 1/2, n_{i(1)} + 3/2)$ for the Jeffreys' Rule prior.

- *Beta* $(n_{i(0)} + 1, n_{i(1)} + 1)$ for the uniform prior.

3. Generate the posterior densities of the $\sigma_i^2|\text{data}$.

- *Inv* $- Gamma[(n_{i(1)} - 1)/2, (n_{i(1)} - 1)\hat{\sigma}_i^2/2]$ for the left-invariant Jeffreys prior.

- *Inv* $- Gamma(n_{i(1)}/2, n_{i(1)}\hat{\sigma}_i^2/2)$ for the Jeffreys' Rule prior.

- *Inv* $- Gamma[(n_{i(1)} - 2)/2, (n_{i(1)} - 2)\hat{\sigma}_i^2/2]$ for the uniform prior.

4. Compute $\gamma$ from Eq. (3).
5. Repeat steps 2–4 5,000 times.
6. Compute the $100(1 - \alpha)\%$ HPD intervals for $\gamma$.
7. Repeat steps 1–6 15,000 times.

**The standard bootstrap method**

Bootstrapping is a type of resampling method that draws samples with replacement from the initial population *Efron (1979)*. According to sample the data $\mathbf{x}_{ij} = (x_{i1}, x_{i2}, \ldots, x_{in_i})$, $i = 1,2$, $j = 1,2,\ldots,n_i$ from a delta-lognormal distribution, let $\mathbf{x}_{ij}^* = (x_{i1}^*, x_{i2}^*, \ldots, x_{in_i}^*)$ be a bootstrap sample from the data. Since $\hat{\delta}_i$ and $\hat{\sigma}_i^2$ are the independent unbiased estimators of $\delta_i$ and $\sigma_i^2$, respectively, the bootstrap estimators of $\delta_i$ and $\sigma_i^2$ are $\hat{\delta}_i^*$ and $\hat{\sigma}_i^{2*}$, respectively. By resampling $K$ bootstrap samples, let $\hat{\gamma}_k^* = \hat{\eta}_{1,k}^* - \hat{\eta}_{2,k}^*$, $k = 1,2,\ldots,K$ be the $k$th bootstrap estimator of $\gamma$. Subsequently, the $100(1 - \alpha)\%$ confidence interval for $\gamma$ using SB is

$$CI_\gamma^{SB} = (\hat{\eta}_1 - \hat{\eta}_2) \pm Z_{1-\frac{\alpha}{2}}S_{\hat{\gamma}^*}^*, \qquad (14)$$

where $S_{\hat{\gamma}^*}^*$ is the standard error of $\hat{\gamma}^*$.

**The algorithm to construct the SB confidence interval**

1. Generate datasets $\mathbf{x}_{ij}$, $i = 1,2$, $j = 1,2,\ldots,n_i$ from a delta-lognormal distribution.
2. Resample samples $\mathbf{x}_{ij}^*$ from $\mathbf{x}_{ij}$.
3. Compute $\hat{\delta}_i^*$ and $\hat{\sigma}_i^{2*}$.
4. Compute $\hat{\gamma}^*$ from Eq. (3).
5. Repeat steps 2–4 3,000 times.
6. Compute the $100(1 - \alpha)\%$ SB confidence interval for $\gamma$.
7. Repeat steps 1–6 15,000 times.

## RESULTS

### The Monte Carlo simulation study

Coverage probabilities and expected lengths were used to compare the performance of the confidence intervals of the proposed methods via Monte Carlo simulation at a nominal confidence level of 0.95. The coverage probabilities that were greater than the nominal confidence level together with the shortest expected lengths were considered as the best. A total of 15,000 replications for each parameter combination were applied for the simulation study involving all of the methods. Moreover, 5,000 duplicates were used for the FGCI and Bayesian methods, and 3,000 resampling samples were used for the bootstrap method. The sample sizes were set as $n_1,n_2 = 25,50,100$; $\mu_1,\mu_2 = 0$; $\delta_1,\delta_2 = 0.2,0.5,0.8$; and $\sigma_1^2, \sigma_2^2 = 0.5, 1.0, 2.0$. Note that in the studies by *Fletcher (2008)* and *Wu & Hsieh (2014)*, the combinations of $n_1,n_2 = 25$; $\delta_1,\delta_2 = 0.2$; and $\sigma_1^2, \sigma_2^2 = 0.5, 1.0, 2.0$ were not considered because the expected non-zero values were less than 10.

The methods to construct confidence intervals for the difference between the independent coefficients of variation of two delta-lognormal distributions were evaluated. The results in Table 1 and Figs. 1–3 show that FGCI was stable and close to the target in terms of coverage probability for almost all cases. For the Bayesian HPD intervals based on the left-invariant Jeffreys prior ($B_{linvj}$), Jeffreys' Rule prior ($B_{jrule}$), and the uniform prior ($B_{uni}$), the coverage probabilities were close to or greater than the target in all cases. In addition, the coverage probabilities of the SB were greater than the target in cases of variances equal to 1.0 and 2.0. However, according to the expected lengths, $B_{jrule}$ mostly had shorter expected lengths than the other method except for a few cases when the sample sizes were large in both groups ($n_1,n_2 = 50,100$) and the variance was equal to 0.5 and 1.0, for which the expected lengths of FGCI were the shortest. Moreover, in cases of $n_1:n_2 = 25:25$, $50:50$, $100:100$ and $\sigma_1^2 : \sigma_2^2 = 1.0:1.0$, $2.0:2.0$, $n_1:n_2 = 25:50$, $50:100$ and $\sigma_1^2 : \sigma_2^2 = 2.0:2.0$, and $n_1:n_2 = 25:100$ and $\sigma_1^2 : \sigma_2^2 = 1.0:2.0$, the SB method had the shortest expected lengths.

### The empirical study

Datasets of rainfall from Thailand were chosen because they usually contain zero values, albeit data containing non-zero values normally follow a lognormal distribution. For rainfall data, *Ananthakrishnan & Soman (1989)* used the normalized rainfall curve

**Table 1 The coverage probabilities and expected lengths of 95% confidence intervals for the difference CVs.**

| $n_1:n_2$ | $\delta_1:\delta_2$ | $\sigma_1^2:\sigma_2^2$ | Coverage probabilities (Expected lengths) | | | | |
|---|---|---|---|---|---|---|---|
| | | | FGCI | $B_{linvj}$ | $B_{jrule}$ | $B_{uni}$ | SB |
| 25:25 | 0.5:0.5 | 0.5:0.5 | 0.9673 | 0.9973 | 0.9966 | 0.9979 | 0.8869 |
| | | | (2.1594) | (2.2567) | (2.0837) | (2.4013) | (1.0054) |
| | | 0.5:1.0 | 0.9559 | 0.9889 | 0.9843 | 0.9913 | 0.8613 |
| | | | (4.6245) | (4.1756) | (3.7971) | (4.5568) | (1.9066) |
| | | 0.5:2.0 | 0.9533 | 0.9627 | 0.9529 | 0.9699 | 0.7995 |
| | | | (33.3988) | (19.2637) | (16.7548) | (22.7213) | (7.2489) |
| | | 1.0:1.0 | 0.9573 | 0.9976 | 0.9957 | 0.9984 | 0.9555 |
| | | | (6.8460) | (6.4599) | (5.7927) | (7.1950) | (2.5935) |
| | | 1.0:2.0 | 0.9530 | 0.9793 | 0.9732 | 0.9845 | 0.8525 |
| | | | (35.1495) | (21.5802) | (18.6556) | (25.3295) | (8.2679) |
| | | 2.0:2.0 | 0.9517 | 0.9983 | 0.9969 | 0.9989 | 0.9913 |
| | | | (68.0825) | (45.4033) | (37.8180) | (54.8612) | (11.6755) |
| | 0.8:0.8 | 0.5:0.5 | 0.9567 | 0.9861 | 0.9827 | 0.9889 | 0.9183 |
| | | | (1.2510) | (1.2894) | (1.2393) | (1.3425) | (0.8089) |
| | | 0.5:1.0 | 0.9523 | 0.9785 | 0.9751 | 0.9836 | 0.9056 |
| | | | (2.3240) | (2.2129) | (2.1182) | (2.3214) | (1.4044) |
| | | 0.5:2.0 | 0.9522 | 0.9608 | 0.9537 | 0.9651 | 0.8502 |
| | | | (9.5540) | (7.3571) | (6.9618) | (7.8340) | (4.5031) |
| | | 1.0:1.0 | 0.9511 | 0.9889 | 0.9848 | 0.9899 | 0.9567 |
| | | | (3.2549) | (3.1689) | (3.0159) | (3.3480) | (1.8576) |
| | | 1.0:2.0 | 0.9529 | 0.9753 | 0.9720 | 0.9796 | 0.8833 |
| | | | (10.1630) | (8.2686) | (7.7973) | (8.8496) | (4.6834) |
| | | 2.0:2.0 | 0.9477 | 0.9940 | 0.9928 | 0.9959 | 0.9917 |
| | | | (3.0485) | (14.5891) | (13.6002) | (15.7798) | (6.8311) |
| 25:50 | 0.5:0.5 | 0.5:0.5 | 0.9624 | 0.9901 | 0.9863 | 0.9921 | 0.8592 |
| | | | (1.7255) | (1.7888) | (1.6832) | (1.8646) | (0.8931) |
| | | 0.5:1.0 | 0.9599 | 0.9909 | 0.9899 | 0.9929 | 0.9269 |
| | | | (2.5974) | (2.6381) | (2.4939) | (2.7539) | (1.5072) |
| | | 0.5:2.0 | 0.9509 | 0.9686 | 0.9642 | 0.9721 | 0.8709 |
| | | | (8.1783) | (7.0822) | (6.7312) | (7.3973) | (4.5482) |
| | | 1.0:1.0 | 0.9549 | 0.9903 | 0.9867 | 0.9921 | 0.9263 |
| | | | (4.9820) | (4.5559) | (4.2090) | (4.9041) | (2.2435) |
| | | 1.0:2.0 | 0.9561 | 0.9877 | 0.9855 | 0.9907 | 0.9216 |
| | | | (10.1142) | (9.3743) | (8.7358) | (10.0319) | (5.0513) |
| | | 2.0:2.0 | 0.9522 | 0.9918 | 0.9900 | 0.9929 | 0.9769 |
| | | | (37.8763) | (26.1295) | (22.8554) | (30.0308) | (9.3159) |
| | 0.8:0.8 | 0.5:0.5 | 0.9528 | 0.9811 | 0.9776 | 0.9841 | 0.9013 |
| | | | (1.0224) | (1.0467) | (1.0148) | (1.0790) | (0.7012) |
| | | 0.5:1.0 | 0.9531 | 0.9805 | 0.9774 | 0.9817 | 0.9360 |
| | | | (1.5016) | (1.5188) | (1.4793) | (1.5628) | (1.0872) |

| | | | Table 1 (continued) | | | | |
|---|---|---|---|---|---|---|---|
| $n_1$:$n_2$ | $\delta_1$:$\delta_2$ | $\sigma_1^2$:$\sigma_2^2$ | Coverage probabilities (Expected lengths) | | | | |
| | | | FGCI | $B_{linvj}$ | $B_{jrule}$ | $B_{uni}$ | SB |
| | | 0.5:2.0 | 0.9533 | 0.9633 | 0.9609 | 0.9651 | 0.9000 |
| | | | (4.1192) | (3.7823) | (3.6962) | (3.8776) | (2.9613) |
| | | 1.0:1.0 | 0.9512 | 0.9812 | 0.9773 | 0.9845 | 0.9398 |
| | | | (2.5113) | (2.4208) | (2.3346) | (2.5203) | (1.6020) |
| | | 1.0:2.0 | 0.9528 | 0.9772 | 0.9745 | 0.9801 | 0.9347 |
| | | | (4.8629) | (4.6740) | (4.5325) | (4.8398) | (3.2412) |
| | | 2.0:2.0 | 0.9508 | 0.9871 | 0.9841 | 0.9898 | 0.9751 |
| | | | (11.2886) | (9.8332) | (9.3589) | (10.4176) | (5.6187) |
| 25:100 | 0.5:0.5 | 0.5:0.5 | 0.9603 | 0.9811 | 0.9749 | 0.9832 | 0.7991 |
| | | | (1.5388) | (1.5690) | (1.4780) | (1.6325) | (0.8035) |
| | | 0.5:1.0 | 0.9587 | 0.9885 | 0.9850 | 0.9899 | 0.9285 |
| | | | (1.9519) | (2.0049) | (1.9077) | (2.0801) | (1.1956) |
| | | 0.5:2.0 | 0.9522 | 0.9769 | 0.9743 | 0.9787 | 0.9147 |
| | | | (4.4100) | (4.2612) | (4.1242) | (4.3739) | (3.1908) |
| | | 1.0:1.0 | 0.9537 | 0.9771 | 0.9696 | 0.9802 | 0.8726 |
| | | | (4.5423) | (3.8597) | (3.5765) | (4.1514) | (2.0158) |
| | | 1.0:2.0 | 0.9530 | 0.9913 | 0.9887 | 0.9926 | 0.9609 |
| | | | (6.5209) | (6.1550) | (5.8006) | (6.5196) | (3.6771) |
| | | 2.0:2.0 | 0.9508 | 0.9789 | 0.9729 | 0.9825 | 0.9217 |
| | | | (36.2420) | (21.8964) | (19.2432) | (25.4336) | (8.1490) |
| | 0.8:0.8 | 0.5:0.5 | 0.9539 | 0.9764 | 0.9719 | 0.9824 | 0.8773 |
| | | | (0.9261) | (0.9326) | (0.9045) | (0.9614) | (0.6381) |
| | | 0.5:1.0 | 0.9533 | 0.9767 | 0.9749 | 0.9793 | 0.9374 |
| | | | (1.1685) | (1.1851) | (1.1561) | (1.2151) | (0.8741) |
| | | 0.5:2.0 | 0.9526 | 0.9672 | 0.9655 | 0.9688 | 0.9323 |
| | | | (2.5254) | (2.4766) | (2.4398) | (2.5174) | (2.0893) |
| | | 1.0:1.0 | 0.9503 | 0.9721 | 0.9685 | 0.9769 | 0.9051 |
| | | | (2.2790) | (2.1192) | (2.0446) | (2.2031) | (1.4489) |
| | | 1.0:2.0 | 0.9537 | 0.9805 | 0.9788 | 0.9827 | 0.9623 |
| | | | (3.3455) | (3.2986) | (3.2136) | (3.3991) | (2.4324) |
| | | 2.0:2.0 | 0.9515 | 0.9759 | 0.9736 | 0.9790 | 0.9254 |
| | | | (10.1643) | (8.1980) | (7.8121) | (8.6649) | (4.9982) |
| 50:50 | 0.2:0.2 | 0.5:0.5 | 0.9687 | 0.9990 | 0.9987 | 0.9993 | 0.8457 |
| | | | (4.3446) | (4.4760) | (3.9330) | (4.8772) | (1.5117) |
| | | 0.5:1.0 | 0.9596 | 0.9938 | 0.9909 | 0.9955 | 0.8318 |
| | | | (11.3576) | (9.2035) | (7.8085) | (10.6120) | (3.0407) |
| | | 0.5:2.0 | 0.9527 | 0.9628 | 0.9497 | 0.9692 | 0.7672 |
| | | | (379.5699) | (68.3314) | (50.3354) | (102.5227) | (13.8885) |
| | | 1.0:1.0 | 0.9589 | 0.9995 | 0.9991 | 0.9997 | 0.9505 |
| | | | (17.8892) | (14.9444) | (12.3859) | (17.8983) | (4.2541) |

(Continued)

| | | | Table 1 (continued) | | | | |
|---|---|---|---|---|---|---|---|
| $n_1{:}n_2$ | $\delta_1{:}\delta_2$ | $\sigma_1^2{:}\sigma_2^2$ | Coverage probabilities (Expected lengths) | | | | |
| | | | FGCI | $B_{linvj}$ | $B_{jrule}$ | $B_{uni}$ | SB |
| | | 1.0:2.0 | 0.9535 | 0.9846 | 0.9763 | 0.9889 | 0.8213 |
| | | | (221.5949) | (102.3997) | (68.1067) | (157.6987) | (13.9408) |
| | | 2.0:2.0 | 0.9527 | 0.9995 | 0.9987 | 0.9996 | 0.9905 |
| | | | (687.4366) | (183.4749) | (127.6218) | (282.3258) | (22.4336) |
| | 0.5:0.5 | 0.5:0.5 | 0.9609 | 0.9921 | 0.9904 | 0.9929 | 0.8980 |
| | | | (1.2063) | (1.3303) | (1.2870) | (1.3520) | (0.7659) |
| | | 0.5:1.0 | 0.9561 | 0.9815 | 0.9789 | 0.9829 | 0.8954 |
| | | | (2.1991) | (2.1818) | (2.1051) | (2.2345) | (1.4150) |
| | | 0.5:2.0 | 0.9528 | 0.9573 | 0.9526 | 0.9599 | 0.8577 |
| | | | (7.8990) | (6.6471) | (6.3821) | (6.8773) | (4.6165) |
| | | 1.0:1.0 | 0.9559 | 0.9888 | 0.9879 | 0.9914 | 0.9531 |
| | | | (3.0060) | (2.9998) | (2.8860) | (3.0909) | (1.8965) |
| | | 1.0:2.0 | 0.9527 | 0.9725 | 0.9679 | 0.9751 | 0.8837 |
| | | | (8.5000) | (7.4233) | (7.1057) | (7.6981) | (4.7694) |
| | | 2.0:2.0 | 0.9514 | 0.9937 | 0.9930 | 0.9954 | 0.9889 |
| | | | (13.1899) | (12.1455) | (11.5553) | (12.6939) | (6.8683) |
| 50:50 | 0.8:0.8 | 0.5:0.5 | 0.9537 | 0.9793 | 0.9785 | 0.9815 | 0.9224 |
| | | | (0.7572) | (0.7973) | (0.7838) | (0.8097) | (0.5878) |
| | | 0.5:1.0 | 0.9521 | 0.9677 | 0.9651 | 0.9700 | 0.9216 |
| | | | (1.2943) | (1.2861) | (1.2634) | (1.3091) | (1.0115) |
| | | 0.5:2.0 | 0.9519 | 0.9565 | 0.9547 | 0.9589 | 0.8909 |
| | | | (4.0145) | (3.5922) | (3.5249) | (3.6685) | (2.9205) |
| | | 1.0:1.0 | 0.9513 | 0.9759 | 0.9736 | 0.9787 | 0.9513 |
| | | | (1.7224) | (1.7283) | (1.6967) | (1.7633) | (1.3158) |
| | | 1.0:2.0 | 0.9521 | 0.9641 | 0.9613 | 0.9670 | 0.9110 |
| | | | (4.2510) | (3.9321) | (3.8539) | (4.0204) | (3.0627) |
| | | 2.0:2.0 | 0.9523 | 0.9865 | 0.9844 | 0.9870 | 0.9845 |
| | | | (6.2427) | (6.0484) | (5.9146) | (6.2018) | (4.2865) |
| 50:100 | 0.2:0.2 | 0.5:0.5 | 0.9640 | 0.9964 | 0.9941 | 0.9964 | 0.8360 |
| | | | (3.2908) | (3.3693) | (3.0608) | (3.5439) | (1.3380) |
| | | 0.5:1.0 | 0.9623 | 0.9969 | 0.9956 | 0.9977 | 0.9038 |
| | | | (4.9483) | (5.0554) | (4.6240) | (5.3501) | (2.3793) |
| | | 0.5:2.0 | 0.9552 | 0.9701 | 0.9655 | 0.9738 | 0.8367 |
| | | | (17.1227) | (14.0971) | (12.9842) | (14.9740) | (7.8643) |
| | | 1.0:1.0 | 0.9587 | 0.9956 | 0.9930 | 0.9968 | 0.9299 |
| | | | (11.9262) | (9.6133) | (8.4150) | (10.8005) | (3.6621) |
| | | 1.0:2.0 | 0.9534 | 0.9909 | 0.9869 | 0.9931 | 0.8973 |
| | | | (22.9239) | (20.1401) | (17.9128) | (22.3911) | (8.7065) |
| | | 2.0:2.0 | 0.9509 | 0.9954 | 0.9919 | 0.9966 | 0.9803 |
| | | | (206.9601) | (100.4277) | (72.9777) | (157.2391) | (18.0215) |

| | | | Table 1 (continued) | | | | |
|---|---|---|---|---|---|---|---|
| $n_1:n_2$ | $\delta_1:\delta_2$ | $\sigma_1^2:\sigma_2^2$ | Coverage probabilities (Expected lengths) | | | | |
| | | | FGCI | $B_{linvj}$ | $B_{jrule}$ | $B_{uni}$ | SB |
| | 0.5:0.5 | 0.5:0.5 | 0.9596 | 0.9859 | 0.9845 | 0.9875 | 0.8779 |
| | | | (1.0067) | (1.1041) | (1.0756) | (1.1170) | (0.6672) |
| | | 0.5:1.0 | 0.9532 | 0.9823 | 0.9816 | 0.9836 | 0.9266 |
| | | | (1.4826) | (1.5699) | (1.5343) | (1.5906) | (1.0971) |
| | | 0.5:2.0 | 0.9503 | 0.9615 | 0.9590 | 0.9637 | 0.9013 |
| | | | (4.0903) | (3.8417) | (3.7690) | (3.8933) | (3.1255) |
| | | 1.0:1.0 | 0.9525 | 0.9801 | 0.9776 | 0.9827 | 0.9359 |
| | | | (2.3930) | (2.3893) | (2.3204) | (2.4426) | (1.6419) |
| | | 1.0:2.0 | 0.9524 | 0.9768 | 0.9756 | 0.9799 | 0.9319 |
| | | | (4.6910) | (4.5889) | (4.4772) | (4.6767) | (3.4066) |
| | | 2.0:2.0 | 0.9501 | 0.9829 | 0.9803 | 0.9847 | 0.9723 |
| | | | (9.6921) | (8.9658) | (8.6364) | (9.2571) | (5.7747) |
| | 0.8:0.8 | 0.5:0.5 | 0.9509 | 0.9723 | 0.9706 | 0.9763 | 0.9179 |
| | | | (0.6389) | (0.6704) | (0.6613) | (0.6788) | (0.5113) |
| | | 0.5:1.0 | 0.9507 | 0.9697 | 0.9678 | 0.9702 | 0.9416 |
| | | | (0.9307) | (0.9511) | (0.9404) | (0.9619) | (0.7815) |
| | | 0.5:2.0 | 0.9533 | 0.9577 | 0.9570 | 0.9595 | 0.9261 |
| | | | (2.3889) | (2.2905) | (2.2691) | (2.3127) | (2.0442) |
| | | 1.0:1.0 | 0.9535 | 0.9701 | 0.9678 | 0.9722 | 0.9411 |
| | | | (1.4198) | (1.4116) | (1.3911) | (1.4335) | (1.1429) |
| | | 1.0:2.0 | 0.9513 | 0.9679 | 0.9667 | 0.9693 | 0.9441 |
| | | | (2.6872) | (2.6428) | (2.6126) | (2.6764) | (2.2221) |
| | | 2.0:2.0 | 0.9505 | 0.9757 | 0.9742 | 0.9783 | 0.9665 |
| | | | (4.9394) | (4.7304) | (4.6503) | (4.8203) | (3.6171) |
| 100:100 | 0.2:0.2 | 0.5:0.5 | 0.9669 | 0.9954 | 0.9950 | 0.9959 | 0.8686 |
| | | | (2.1090) | (2.3784) | (2.2658) | (2.4068) | (1.1520) |
| | | 0.5:1.0 | 0.9562 | 0.9871 | 0.9826 | 0.9881 | 0.8686 |
| | | | (3.9894) | (3.9890) | (3.7803) | (4.0876) | (2.2483) |
| | | 0.5:2.0 | 0.9513 | 0.9582 | 0.9510 | 0.9615 | 0.8265 |
| | | | (16.4620) | (12.9759) | (12.1641) | (13.5255) | (7.9611) |
| | | 1.0:1.0 | 0.9571 | 0.9943 | 0.9941 | 0.9961 | 0.9517 |
| | | | (5.5425) | (5.6080) | (5.2895) | (5.7983) | (3.0592) |
| | | 1.0:2.0 | 0.9538 | 0.9755 | 0.9709 | 0.9775 | 0.8632 |
| | | | (17.6427) | (14.6645) | (13.7125) | (15.3659) | (8.2895) |
| | | 2.0:2.0 | 0.9523 | 0.9963 | 0.9953 | 0.9971 | 0.9896 |
| | | | (27.9500) | (25.3838) | (23.5151) | (26.9001) | (12.1880) |
| | 0.5:0.5 | 0.5:0.5 | 0.9564 | 0.9865 | 0.9853 | 0.9874 | 0.9045 |
| | | | (0.7720) | (0.8662) | (0.8530) | (0.8708) | (0.5607) |
| | | 0.5:1.0 | 0.9570 | 0.9731 | 0.9703 | 0.9740 | 0.9140 |
| | | | (1.3204) | (1.3613) | (1.3402) | (1.3733) | (1.0281) |

(Continued)

| Table 1 (continued) | | | | | | | |
|---|---|---|---|---|---|---|---|
| $n_1{:}n_2$ | $\delta_1{:}\delta_2$ | $\sigma_1^2{:}\sigma_2^2$ | Coverage probabilities (Expected lengths) | | | | |
| | | | FGCI | $B_{linvj}$ | $B_{jrule}$ | $B_{uni}$ | SB |
| | | 0.5:2.0 | 0.9503 | 0.9549 | 0.9525 | 0.9559 | 0.8907 |
| | | | (4.0361) | (3.6979) | (3.6367) | (3.7426) | (3.1053) |
| | | 1.0:1.0 | 0.9518 | 0.9779 | 0.9761 | 0.9792 | 0.9519 |
| | | | (1.7362) | (1.7898) | (1.7605) | (1.8090) | (1.3591) |
| | | 1.0:2.0 | 0.9511 | 0.9643 | 0.9619 | 0.9654 | 0.9125 |
| | | | (4.2295) | (4.0091) | (3.9418) | (4.0614) | (3.2737) |
| | | 2.0:2.0 | 0.9505 | 0.9833 | 0.9811 | 0.9843 | 0.9809 |
| | | | (6.1135) | (6.0198) | (5.9072) | (6.1094) | (4.5458) |
| | 0.8:0.8 | 0.5:0.5 | 0.9531 | 0.9719 | 0.9719 | 0.9734 | 0.9296 |
| | | | (0.5013) | (0.5311) | (0.5269) | (0.5348) | (0.4237) |
| | | 0.5:1.0 | 0.9511 | 0.9643 | 0.9625 | 0.9653 | 0.9302 |
| | | | (0.8306) | (0.8394) | (0.8328) | (0.8460) | (0.7232) |
| | | 0.5:2.0 | 0.9525 | 0.9505 | 0.9495 | 0.9521 | 0.9194 |
| | | | (2.3561) | (2.2276) | (2.2088) | (2.2473) | (2.0198) |
| | | 1.0:1.0 | 0.9529 | 0.9641 | 0.9618 | 0.9653 | 0.9511 |
| | | | (1.0792) | (1.0925) | (1.0835) | (1.1015) | (0.9386) |
| | | 1.0:2.0 | 0.9534 | 0.9614 | 0.9606 | 0.9637 | 0.9299 |
| | | | (2.4814) | (2.3968) | (2.3755) | (2.4182) | (2.1126) |
| | | 2.0:2.0 | 0.9521 | 0.9729 | 0.9710 | 0.9742 | 0.9747 |
| | | | (3.4679) | (3.4336) | (3.4027) | (3.4682) | (2.8902) |

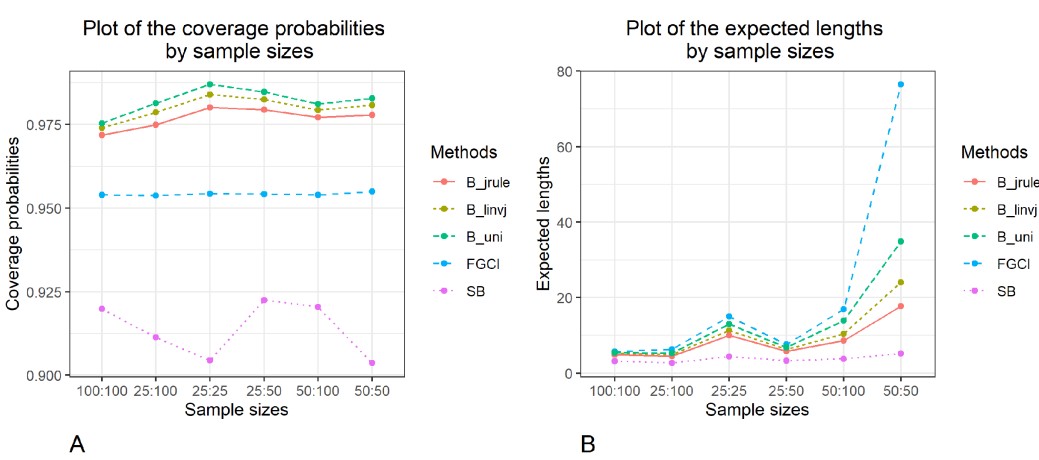

**Figure 1 Graphs to compare the performance of the proposed methods in terms of (A) coverage probability (B) expected length with varying sample size.**

(NRC) to describe the relationship between the accumulated percentage of the rain amount and the number of rain days in a rainfall series. Their results indicate that the coefficient of variation of the rainfall datasets can be used in the unique determination of

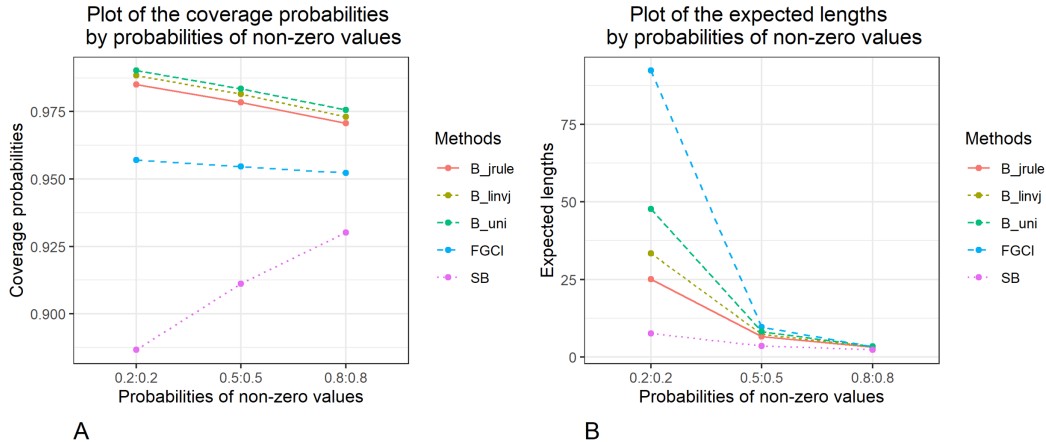

**Figure 2 Graphs to compare the performance of the proposed methods in terms of (A) coverage probability (B) expected length with varying probabilities of non-zero values.**

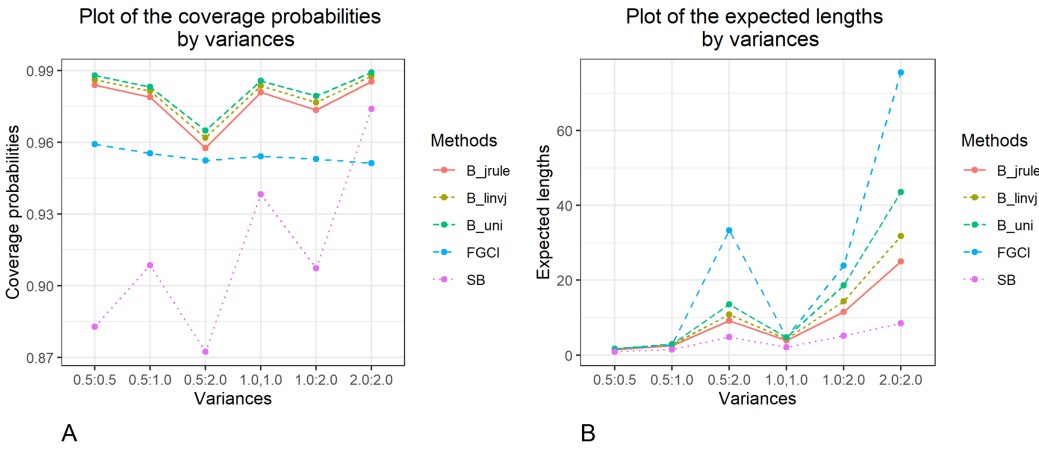

**Figure 3 Graphs to compare the performance of the proposed methods in terms of (A) coverage probability (B) expected length with varying variances.**

the NRC. Moreover, *Shimizu (1993)* introduced a probability model for a combination of bivariate and lognormal distributions to represent rainfall data. The author used monthly rainfall data from Jana and Ranod, Songkhla, Thailand from 2008 to 2017 to illustrate confidence intervals for the difference between coefficients of variation from two areas.

Songkhla is located on the east coast of southern Thailand and is somewhat rainy due to the influences of the southwest monsoon coming from the Indian Ocean and the northeast monsoon coming from the Gulf of Thailand. This area has a lot of rain from May to December, which decreases from January to April (the datasets were collected by the Southern Meteorological Center (East Coast)). These datasets included both positive and true-zero observations. The positive values for each area create skewness, as shown in Fig. 4, and thus their distributions were subjected to Akaike information criterion (AIC) analyses. The AIC values according to normal, Cauchy, lognormal, exponential, and gamma distributions in Jana were 1421.5050, 1355.5600, 1279.9710, 1281.8810, and

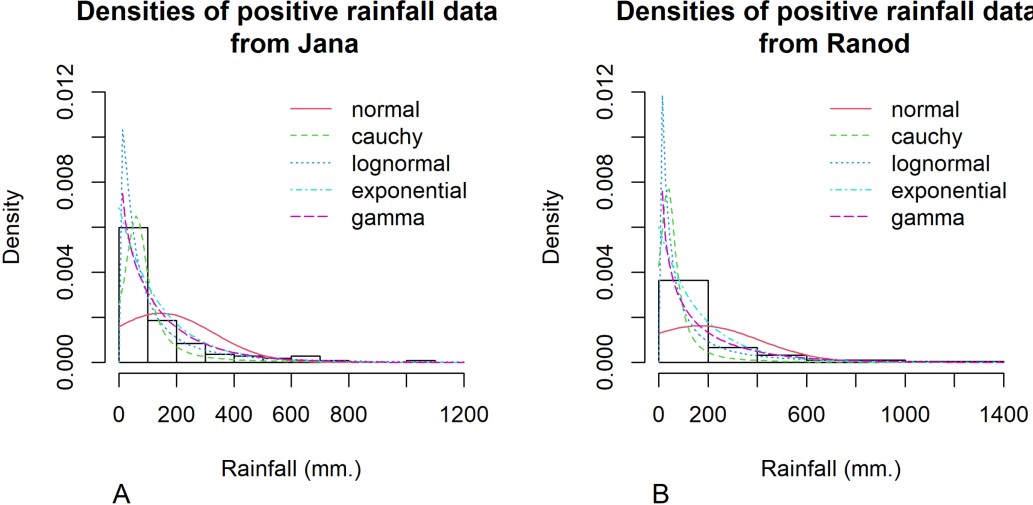

**Figure 4 Histogram and theoretical densities of the positive monthly rainfall data from (A) Jana (B) Ranod, Songkhla, Thailand from 2008 to 2017.**

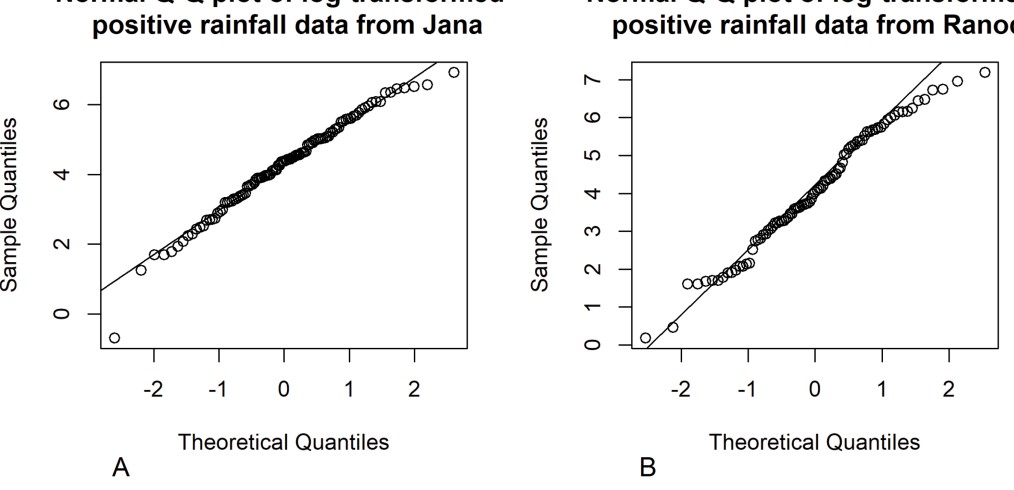

**Figure 5 Normal Q–Q plots of the log-transformed positive monthly rainfall data from (A) Jana (B) Ranod, Songkhla, Thailand from 2008 to 2017.**

1280.7610 and in Ranod were 1234.7680, 1155.3230, 1063.9270, 1090.1390, and 1073.1520, respectively. The AIC values indicating a lognormal distribution were less than the others, and so the positive datasets from Jana and Ranod are lognormal distributions. To confirm AIC results, the normality plots of the log-transformation of the monthly rainfall data from both areas in Figs. 5 and 6 indicate that both rainfall series are lognormal distributions. Moreover, the true-zero values from Jana and Ranod are binomial distributions. Therefore, the distributions of the monthly rainfall series from Jana and Ranod are delta-lognormal. The summary statistics for Jana are $n_1 = 120$, $\hat{\delta}_1 = 0.8917$, $\hat{\mu}_1 = 4.2556$, $\hat{\sigma}_1^2 = 1.7953$, and $\hat{\eta}_1 = 0.3149$ and for Ranod are $n_2 = 120$, $\hat{\delta}_2 = 0.7417$, $\hat{\mu}_2 = 4.0846$, $\hat{\sigma}_2^2 = 2.4928$, and $\hat{\eta}_2 = 0.3865$. The difference between $\hat{\eta}_1$ and $\hat{\eta}_2$ is $\gamma = -0.0716$. The 95% confidence intervals for FGCI and SB are (−4.0492, −0.0558),

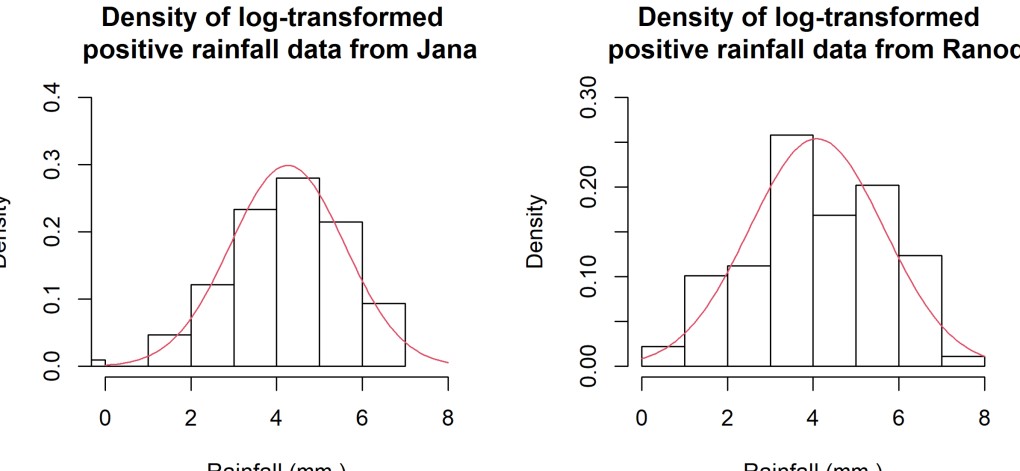

**Figure 6 Histogram and theoretical density of the log-transformed positive monthly rainfall data from (A) Jana (B) Ranod, Songkhla, Thailand from 2008 to 2017.**

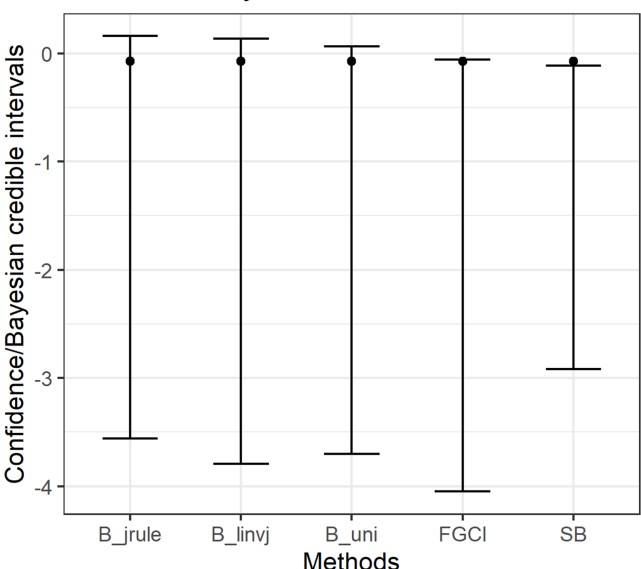

**Figure 7 The 95% confidence intervals and Bayesian credible intervals for the difference between coefficients of variation of the monthly rainfall data from Jana and Ranod, Songkhla, Thailand from 2008 to 2017.**

and ($-2.9163$, $-0.1118$) with interval lengths of $3.9934$ and $2.8045$, respectively. Similarly, the 95% Bayesian HPD intervals using the left-invariant Jeffreys, Jeffreys' Rule, and uniform priors are ($-3.7924$, $0.1359$), ($-3.5602$, $0.1619$), and ($-3.7008$, $0.0646$) with interval lengths $3.9283$, $3.7221$, and $3.7654$, respectively. These intervals are shown in Fig. 7. The Bayesian method using the Jeffreys' Rule prior outperformed the others in terms of the coverage probability and interval length. Therefore, these results are in

accordance with those from the simulation studies when the variance is large. Furthermore, the results for the Bayesian method using the Jeffreys' Rule prior demonstrate that there is not a difference in the rainfall intensity between the areas.

## DISCUSSION

Herein, Bayesian and SB methods are proposed to construct confidence intervals for the difference between delta-lognormal coefficients of variation and then compared with FGCI recommended by *Yosboonruang, Niwitpong & Niwitpong (2019a)*. It was found that the coverage probabilities of FGCI were more consistent with the target than the Bayesian and SB methods. The coverage probabilities of the Bayesian method were greater than the nominal confidence level and mostly close to 1.00, which suggests overestimation. Nevertheless, the expected lengths of the Bayesian method using the Jeffreys' Rule prior were shorter than FGCI in almost every case. This is due to the criterion that the posterior density values at the lower and upper limits are equal, which was applied for constructing the confidence intervals of the Bayesian methods. Moreover, in case of small variances, it is notable that the expected lengths of the confidence intervals were sufficiently narrow. This indicated that FGCI and the Bayesian methods can be efficiently used to construct the confidence intervals. Furthermore, the coverage probabilities of the SB method were greater than the nominal confidence level only for the large variance cases, although remarkably, it supplied the shortest expected lengths. However, these three methods required a large amount of computing to obtain the interval estimates due to FGCI must be calculated GFQ for parameters of interest ($\delta_i$ and $\sigma_i^2$) and the Bayesian method must be obtained the posterior densities of $\tilde{\delta}_i$ and $\sigma_i^2$. In addition, SB method have to resample bootstrap samples for computing the estimators of $\delta_i$ and $\sigma_i^2$ which takes more time than FGCI and the Bayesian methods. The results using the two rainfall data series were matched with the simulation, with the Bayesian method using the Jeffreys' Rule prior demonstrating the difference between their coefficients of variation much better than the others.

## CONCLUSIONS

In this study, the three concepts: FGCI, Bayesian, and SB methods were used to construct five confidence intervals for the difference between two independent coefficients of variation of a delta-lognormal distribution. Of these, the Bayesian method was used to construct three confidence intervals using the left-invariant Jeffreys, Jeffreys' Rule, and uniform priors under HPD intervals. Other confidence intervals based on the SB method and FGCI were also used.

The results of the simulation studies indicate that the performance of the Bayesian HPD based on the Jeffreys' Rule prior performed the best in almost all cases. Although the coverage probabilities were close to 1.00 for all of the priors, the expected lengths of the Jeffreys' Rule prior were shorter for the confidence intervals of the difference between the coefficients of variation of two delta-lognormal distributions in almost all cases. Moreover, FGCI is appropriate for a large sample size together with small variance while the SB method is suggested for a large variance. Furthermore, a comparison of the simulation

results and using sets of real data indicates that the Bayesian method using the Jeffreys'
Rule prior can be recommended for constructing the confidence intervals for the difference
between two independent coefficients of variation of a delta-lognormal distribution.

### Funding
This research was funded by King Mongkut's University of Technology North Bangkok.
Grant No. KMUTNB-60-ART-090. The funders had no role in study design, data
collection and analysis, decision to publish, or preparation of the manuscript.

### Grant Disclosures
The following grant information was disclosed by the authors:
King Mongkut's University of Technology North Bangkok: KMUTNB-60-ART-090.

### Competing Interests
The authors declare that they have no competing interests.

### Author Contributions
- Noppadon Yosboonruang conceived and designed the experiments, performed the
  experiments, analyzed the data, prepared figures and/or tables, authored or reviewed
  drafts of the paper, and approved the final draft.
- Sa-Aat Niwitpong conceived and designed the experiments, analyzed the data, prepared
  figures and/or tables, authored or reviewed drafts of the paper, and approved the final
  draft.
- Suparat Niwitpong conceived and designed the experiments, performed the
  experiments, analyzed the data, authored or reviewed drafts of the paper, and approved
  the final draft.

### Data Availability
  The raw data and code are available in the Supplemental Files.

### Supplemental Information
Supplemental information for this article can be found online at http://dx.doi.org/10.7717/
peerj.9662#supplemental-information.

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
