# Peer review of "The Bayesian confidence intervals for measuring the difference between dispersions of rainfall in Thailand"

_PeerJ, doi:10.7717/peerj.9662_

## Round 0.1 · original submission · Minor Revisions

Two favorable reviews have been received, both identifying a number of minor changes required. Note that there is currently there is one review outstanding for more than 2 weeks, but I do not wish to delay the processing further.

Please provide a detailed summary of the rainfall data used in the study, including descriptive summary statistic, provide additional plots as requested and please enlarge and clarify the figures.

Some additional discussion and/ or detail is also required – in particular:
- A justification of the methodology for confidence estimation, given the relatively small samples, is needed.
- The impact of small variations in the data on the estimation of confidence intervals should be detailed.
- Discuss the amount of computing required for the estimation.
- In addition, some specific clarifications as requested by reviewer 2.

Reviewer 1 ·

Basic reporting

WELL REPORTED

Experimental design

RELIABLE

Validity of the findings

STATISTICALLY OK.

Additional comments

Review of article titled "The Bayesian confidence intervals for measuring the difference between dispersion's of rainfall in Thailand". The review congratulates the authors for pointing out a firm understanding of what sorts of inferences confidence interval theory does, and does not, allow which is critical to illustrate the variability in precipitation. However, there exists caution against relying upon confidence interval theory to justify interval estimates, and suggest that other theories of interval estimation should be used instead.
The following are some of the points the authors need to address for possible acceptance of the manuscript.
1. The authors nowhere provide descriptive statistics of rainfall data used in the study.
2. Confidence interval evaluation by any method can be unreliable for small samples (<10000) as in your situation. Can authors justify?
3. Likelihoods, confidence intervals, and Bayesian credible intervals should be plotted.
4. Discuss on dependence of the interval on small variations in the data and the large amount of computing required to obtain the interval estimates.

Reviewer 2 ·

Basic reporting

This is a nicely written paper with a clear message. It compares different approaches for confidence interval estiamtion in the case of difference of two coefficients of variation. The Bayesian approach using Jeffreys' prior as well as the Fiducial Generalized CI comes out beneficial.

Experimental design

not applicable here.

Validity of the findings

The findings are real and substantiated.

Additional comments

There are a few points I wish to make.

l108: " since the coverage probabilities do not cover the target in a few cases" does not make sense to me. CIs corver the target or not, but coverage probabilities?

l126 and what follows: it would be good to say how the coefficient of variation is actually defined. In fact, on line 65 it is said that the CV "is the proportion of the standard deviation to the mean". I do not think that this is right.

(10) what is the Fisher information here? What does | | mean here? Determinant or absolute value or both? Can you motivate why this is called left-invariant Jeffreys' prior?

Fig1 -3 are quite small and you will score more sales here if you produce better graphs. Clearly, make the bench mark line visible. It would be good to incorporate some text into the figures to make these plots more illustrative.

---

## Round 0.2 · accepted · Accept

Thank you for the careful changes to your manuscript. I am happy to recommend that this is accepted for publication in its current form.

Reviewer 1 ·

Basic reporting

I am convinced by the revised version of the manuscript. It's now of acceptable form.

Experimental design

Relevant and satisfactory

Validity of the findings

Reliable and satisfactory

Additional comments

I recommend for acceptance of the manuscript in current form.

Reviewer 2 ·

Basic reporting

Revision is fine.

Experimental design

No issues here.

Validity of the findings

Ok.

Additional comments

Thank you for a careful revision.